# Disentangling the Long-Term Effects of Recommendations on User Consumption Patterns

## ABSTRACT

Recommendation algorithms play a pivotal role in shaping our media choices, which makes it crucial to comprehend their long-term impact on user behavior. These algorithms are often linked to two critical outcomes: homogenization, wherein users consume similar content despite disparate underlying preferences, and the filter bubble effect, wherein individuals with differing preferences only consume content aligned with their preferences (without much overlap with other users). Prior research assumes a trade-off between homogenization and filter bubble effects and then shows that personalized recommendations mitigate filter bubbles by fostering homogenization. However, because of this assumption of a tradeoff between these two effects, prior work cannot develop a more nuanced view of how recommendation systems may independently impact homogenization and filter bubble effects. We develop a more refined definition of homogenization and the filter bubble effect by decomposing them into two key metrics: how different the average consumption is between users (inter-user diversity) and how varied an individual's consumption is (intra-user diversity). We then use a novel agent-based simulation framework that enables a holistic view of the impact of recommendation systems on homogenization and filter bubble effects. Our simulations show that traditional recommendation algorithms (based on past behavior) mainly reduce filter bubbles by affecting inter-user diversity without significantly impacting intra-user diversity. Building on these findings, we introduce two new recommendation algorithms in our simulation model that consider both types of diversity to create more effective recommendation systems.

### ACM Reference Format:
Anonymous Author(s). 2023. Disentangling the Long-Term Effects of Recommendations on User Consumption Patterns. In *Proceedings of ACM Conference (Conference'17)*. ACM, New York, NY, USA, 11 pages. https://doi.org/XXXXXXX.XXXXXXX

## 1 INTRODUCTION

With the advent of the Internet, much of our social interaction and entertainment has moved online, dispersed across various platforms that each curate their own content. Recommendation algorithms help us navigate these content collections, influencing our choices by providing context. However, lingering questions exist about the effects of these algorithms on our media consumption and social

*Conference'17, July 2017, Washington, DC, USA*
© 2023 Association for Computing Machinery.
ACM ISBN 978-x-xxxx-xxxx-x/YY/MM...$15.00
https://doi.org/XXXXXXX.XXXXXXX

behavior. Previous research has examined their role in fostering homophilous communities [10], amplifying a *rich-get-richer* effect in online social ties [19], and potential bias against minority users [12].

This paper aims to deepen our understanding of two key phenomena often linked to recommendation algorithms: *homogenization* and *filter bubbles*. Past studies (e.g., Nguyen et al. [15], Aridor et al. [1]) indicate that personalized recommendations based on past consumption can mitigate filter bubble effects, but they do so at the expense of homogenizing the audience. These findings, however, only look at how homogeneous agents are in terms of the average item consumed by each agent, and do not examine the diversity of consumption of individual users. Thus, the question remains: do these algorithms diversify or homogenize the set of items any particular individual consumers? The answer to this question has important implications for recommendation algorithm design.

Given the relative lack of control over confounding factors when using observational data, we explore these questions through a simulation study using agent-based modeling. We start by proposing a novel simulation model consisting of users and items. Each item has a quality and a genre, both represented via real numbers. On the other hand, each user has an underlying preference for what genre of item they like the most, also represented via a real number Quality indicates how universally desirable the item is, while the genre of an item impacts different users differently as users prefer to consume items nearer their genre preference. When deciding which item to consume, users estimate and maximize item utility according to a set of available signals, including a recommendation signal provided by the system. In our study, we simulate seven recommendation algorithms: four of these act as idealized baselines, while the remaining three are based on past consumption.

Our first contribution is to disentangle the effects of recommendation algorithms on two types of diversity: **inter-user diversity**, which measures how the mean of individual consumption varies across users, and **intra-user diversity**, which measures how diverse individual consumption is on average. This insight leads us to operationalize a new definition of the filter bubble effect as a ratio between inter-user and intra-user diversity. The intuition behind our definition is that a weak filter bubble effect exists when all users consume the same blockbuster items (i.e., low inter-user diversity), but also when each individual user consume items from a wide range of genre and are not just confined to their own preference (i.e. high intra-user diversity). Results from our simulations show that the past consumption-based recommendations alleviate the filter bubble effect only by homogenizing the population towards blockbuster items and reducing inter-user diversity, without significantly affecting intra-user diversity.

As our second contribution, we propose two novel recommendation ideas: **binned consumption recommendation** and **skewed top pick recommendation**, inspired by the insight that understanding the dynamics between homogenization and filter bubbles

requires examining both inter-user and intra-user diversity. Binned consumption recommendation recommends the set of most consumed items in each genre, therefore recommending a curated set of items and eliminating bias towards blockbuster items. This recommendation alleviates the filter bubble effect by not only decreasing inter-user diversity, but also significantly increasing intra-user diversity. On the other hand, skewed top pick recommendation prioritizes exposure to more niche items. Rather than alleviating the filter bubble effect, this recommendation focuses on simultaneously increasing inter-user and intra-user diversity.

These novel recommendation algorithms are, on the surface, very similar to prior work which intentionally recommends a diverse slate of items to users [4, 11, 14]. However, there are also important differences: we are studying which items are consumed rather than which items are recommended. This difference is impacted by the way that agents make use of the system's recommendations. This in turn is subtly affected by the entirety of information the system is providing and the other information available to the agent.

The rest of this paper is organized as follows: section 3.2 describes our simulation model. Section 4 delves deeper into our definitions of homogeneity and the filter bubble effect in terms of inter-user and intra-user diversity. Section 5 describes the recommendation algorithms we simulate. Section 7 presents our simulation results, providing a more comprehensive picture of the dynamics between homogenization and filter bubble effects (section 7). Section 8 describes two novel recommendation ideas that stem from the insights we received from our simulation results.

## 2 RELATED WORK

Our work combines multiple streams of research on recommendation algorithms. In particular, our research builds on past literature investigating the role of recommendation algorithms in reducing inter-user diversity through homogenization, causing filter bubbles, and the possible interplay between these two phenomena.

• **Homogenization and inter-user diversity:** The existing literature strongly supports a connection between recommendation algorithms and homogenization, largely attributing this to a popularity bias or feedback loop that continually directs users toward a common set of popular items —often referred to as blockbusters. For instance, Salganik et al. [18] used an experimental method allowing participants to listen to a song and decide to download it based on its popularity. The study revealed a widening disparity in song success, signaling a popularity bias. Fleder and Hosanagar [7] employed a 2D simulation model of consumers and items, showing through the Gini coefficient that sales diversity diminishes with collaborative filtering-based recommendations. Similarly, Chaney et al. [5] used a more complex simulation where consumption choices are deterministic, based on both recommendation ranking and personal utility signals. Their findings indicate that user consumption overlap, and thus homogenization, increases over time. Mansoury et al. [13] adopted a hybrid simulation using a real movie dataset and leveraged KL divergence to demonstrate convergence in genre distributions among users. Across these studies, it is evident that recommendation algorithms reinforce the popularity of already well-consumed items, pushing the general population toward these choices and perpetuating the cycle.

• **Filter bubbles:** The existence of filter bubbles is far more contentious than that of homogenization. Eli Pariser first introduced the term "Filter Bubble" in 2011 to describe how personalization could limit exposure to content that diverges from user preferences [17]. However, he didn't provide a definitive framework, resulting in an ongoing debate marked by an absence of a universally accepted, operational definition [3].

One approach to defining filter bubbles is to adopt Pariser's original concept of literal "bubbles" or "filters" that fully restrict exposure to non-conforming content. Counterarguments suggest that recommendation algorithms actually expand user horizons. Flaxman et al. [6], for instance, found that recommendation algorithms expose consumers to more diverse news than they would find independently. Similarly, Hosanagar et al. [9] discovered that music recommendations on iTunes do not increase the distance between user clusters, indicating a broadening of musical exposure. Alternatively, filter bubbles can be conceptualized as focused exposure to content that aligns with user preferences. While algorithms may introduce some diverse content, they predominantly amplify existing preferences. O'Callaghan et al. [16] found that top-$K$ related YouTube channels often mirror the political orientation of the original channel, suggesting concentrated exposure. Bakshy et al. [2] revealed a 15% reduction in exposure to conflicting viewpoints on Facebook due to news feed filtering. Geschke et al. [8] further bolstered this view using agent-based modeling to show that social and technological factors enhance naturally occuring filter bubble effects.

In this paper, we adopt a nuanced perspective that eschews the notion of a literal "bubble." Instead, we define the filter bubble effect on a continuum, and it intensifies when individuals with different preferences increasingly consume disparate content or items.

• **Dynamics between homogenization and filter bubbles:** There is limited prior work that directly examines the interplay between homogenization and filter bubbles. To our knowledge, only two studies—by Nguyen et al. [15] and Aridor et al. [1]—address this trade-off. Nguyen et al. work with the MovieLens dataset, where users get personalized recommendations and rate movies post-viewing. Aridor et al. employ a simulation where items have both social and user-specific valuations, with recommendations tailored to the latter. They operationalize the filter bubble effect as focused exposure and consumption by users. On the other hand, they operationalize homogenization as increasing overlap between the consumption of different users, the opposite of how we interpret the filter bubble effect. They then show that personalized recommendations their measure of homogeneity and conclude that it also decreases filter bubbles.

Our research enriches this body of work by revealing that the filter bubble effect can be disentangled into its impact on both inter-user and intra-user diversity. We argue that the trade-off between homogenization and filter bubbles is not merely the opposite as previously thought. Specifically, recommendations can not only impact inter-user diversity but also augment intra-user diversity—a facet unaccounted for in prior studies.

# 3 RESEARCH DESIGN

## 3.1 Research question

As previously stated, we are interested in uncovering a more complete picture of the dynamics between the filter bubble and homogenization effects of recommendation algorithms. Hence, we seek to answer the following research question in this paper:

*Can we explain the dynamics between homogenization and filter bubble effects of recommendations beyond a simple trade-off by considering both inter-user and intra-user diversity?*

## 3.2 Simulation model

The core building blocks of our simulated world $\mathcal{W}$ are $m$ users and $n$ items. Each user $j = 1, 2, \ldots, m$ has an associated genre preference $p_j$ drawn independently from some distribution $\mathcal{P}$, i.e.

$$p_j \sim_R \mathcal{P} \qquad \forall j = 1, 2, \ldots, m$$

On the other hand, each item $i$ in this world has some inherent quality $q_i$ drawn independently from distribution $\mathcal{Q}$, as well as some genre attribute $g_i$ drawn independently from distribution $\mathcal{G}$.

$$q_i \sim_R \mathcal{Q}, \qquad g_i \sim_R \mathcal{G} \qquad \forall i = 1, 2, \ldots, n$$

We choose to use real values for user preferences and item genres because it allows us to distinguish and observe niche users and items without adding additional complexity to our model. Specifically, users with preferences situated away from the modes of the preference distribution are considered niche. Similarly, items with genres situated away from the modes of the genre distribution are considered niche.

The progression of time $t$ in our simulated world $\mathcal{W}$ is discrete, and continues for $T$ rounds. Initially, the world consists of $k_{init}$ items. At each of the $T$ discrete rounds, $k_{new}$ items are added to the world. Therefore, $n = k_{init} + T \cdot k_{new}$.

User utility in our model consists of two components: a shared quality component corresponding to the quality $q_i$ of item $i$, and an affinity component corresponding to the loss due to misalignment between the genre $g_i$ of item $i$ and the preference $p_j$ of user $j$.

**Definition.** *The **true utility** received by user $j$ by consuming item $i$ is $U(j, i) = q_i - |p_j - g_i|$.*

In each round, each user consumes exactly one item, at which point the said item becomes unavailable to them for future consumption. Following convention, we model users as utility maximizers: in each round, a user $j$ attempts to choose the item $i$ that would yield the maximum utility for them from the set of items they have yet to consume.

However, users do not know an item's true quality or true genre, and therefore cannot directly compute its true utility. Instead, in each round $t$, each user estimates the utility of each previously unconsumed item as a function $\mathcal{F}$ of the following three signals available to them:

(1) A private signal $q_i^j$, which is a noisy personal estimate of the quality of item $i$ obtained by adding some noise $\xi_i^j$ drawn from distribution $\mathcal{N}_{qual}$ to the true quality $q_i$:

$$q_i^j = q_i + \xi_i^j, \qquad \xi_i^j \sim_R \mathcal{N}_{qual}$$

(2) The perceived distance between their preference and the item genre. User $j$ has a noisy personal estimate $g_i^j$ of the true genre of item $i$, obtained by adding some noise $\delta_i^j$ drawn from distribution $\mathcal{N}_{genre}$ to the true genre $g_i$:

$$g_i^j = g_i + \delta_i^j, \qquad \delta_i^j \sim_R \mathcal{N}_{genre}$$

The signal used by user $j$ for estimating utility is $|p_j - g_i^j|$.

(3) Recommendation $\mathbf{r}_i^j(t)$ provided by the system consisting of one or more pieces of information about the item, such as consumption numbers etc. (i.e., $\mathbf{r}_i^j(t)$ is a vector of real numbers)

In other words, in each round $t$, each user $j$ chooses available item $i$ that maximizes the estimated utility

$$\hat{U}(j, i, t) = \mathcal{F}\left(q_i^j, |p_j - g_i^j|, \mathbf{r}_i^j(t)\right)$$

The implication of designing user utility in this manner is as follows: the shared quality component means that a user can learn something about each item from all other users. However, the affinity component means that a user is intuitively best informed by other similar users. Therefore, if a user only listens to similar users, they fail to learn as much as possible about the quality component. But if they listen too much to the global consensus, they fail to learn as much as possible about the affinity component.

Overall, the world $\mathcal{W}$ in our simulation framework can be defined via the collection of distributions used in order to generate the user and item properties:

$$\mathcal{W} = (\mathcal{P}, \mathcal{Q}, \mathcal{G}, \mathcal{N}_{qual}, \mathcal{N}_{genre})$$

*3.2.1 Utility estimation by users.* While the exact nature of the estimator $\mathcal{F}$ is unknown, we can use a machine learning model as a suitable replacement for it in our simulation. This model would take private signal $q_i^j$, perceived distance between preference and genre $|p_j - g_i^j|$ and recommendation $\mathbf{r}_i^j(t)$ as features. In other words, the feature vectors for users $j$ and items $i$ in round $t$ for this ML model are given by

$$\mathbf{x}_{ji}(t) = \left[q_i^j, |p_j - g_i^j|, \mathbf{r}_i^j(t)\right]$$

For simplicity, we replace $\mathcal{F}$ with a linear regression model[1]: at each step, user $j$ chooses item $i$ to consume based on $\hat{U}(j, i)$ estimated via

$$\hat{U}(j, i, t) = w_0 + \mathbf{w}^\top \mathbf{x}_{ji}(t) \qquad \forall j, i$$

We assume that the coefficients $w_0, \mathbf{w}$ are not user-specific. The details about how we learn this regression model in order to run our simulation are provided in section 6.

# 4 MEASURING RECOMMENDATION EFFECTS

In this section, we describe in detail the measures we use to explain and justify our primary contribution: ***inter-user diversity***, ***intra-user diversity***, and a new definition of the filter bubble effect in terms of these two. However, before we can describe these measures, we need to define some preliminary concepts.

We start with the consumed set of a given user, simply the ordered collection of items they consume.

---

[1]We experimented with using a more complex multi-layer perceptron neural network model, but the results were qualitatively the same.

**Definition.** *The **consumed set** of a user $j$ at time $t$ is defined to be an ordered collection $C_j(t)$ of items consumed by the user prior to round $t$, i.e. $C_j(t) = (c_j^1, c_j^2, \ldots, c_j^{t-1})$.*

For brevity, we omit $t$ from the notation and assume that $C_j$ represents consumption of user $j$ after the final round, i.e., after round $T$, unless stated otherwise.

Given the items consumed by a user, we can take the mean of the genres of these items as an indicator of what general genre of items they consume. We define this as the *mean consumed genre*.

**Definition.** *The **mean consumed genre** of a user $j$, $\mu_j = \frac{1}{T} \sum_{i \in C_j} g_i$, is the mean of the genres of the items consumed by the user $C_j$.*

On the other hand, we can take the variance of the genres of the items consumed by the user to indicate of how broad their consumption is. We define this as the *consumed genre variance*.

**Definition.** *The **consumed genre variance** of a user $j$, $\sigma_j^2 = \mathrm{Var}\left[\{g_i | i \in C_j\}\right]$, is the variance of the genres of the items consumed by the user $C_j$.*

### 4.1 Inter-user and intra-user diversity

Inter-user diversity measures how diverse average individual consumption is across all users. We previously defined mean consumed genre as a representation of the average consumption of an individual user. Therefore, we can measure inter-user diversity by taking the variance of mean consumed genre across all users. Formally,

**Definition.** *Given a group of users $\mathcal{U}$ with their respective consumed sets, we define their **inter-user diversity** of consumption as the variance of their individual mean consumed genre, mathematically given as $\mathrm{Var}_{j \in \mathcal{U}}\left[\mu_j\right]$.*

On the other hand, intra-user diversity measures how broad the consumption of a random individual user is, indicating whether individual users are consuming from a very narrow genre range or if they are exposed to many different genres. We previously defined consumed genre variance as a measure of how broad the consumption of an individual is. Therefore, we can measure intra-user diversity by taking the mean of consumed genre variance across all users. Formally,

**Definition.** *Given a group of users $\mathcal{U}$ with their respective consumed sets, we define their **intra-user diversity** of consumption as the mean of their individual consumed genre variance, mathematically given as $\mathbb{E}_{j \sim \mathcal{U}}\left[\sigma_j^2\right]$.*

### 4.2 Homogenization and filter bubble effect

At the heart of this paper is the argument that we need to examine both inter-user and intra-user diversity to fully understand the role of recommendations in the dynamics between homogenization and filter bubbles, which necessitates expressing homogeneity and the filter bubble effect in terms these two types of diversity.

Our high-level interpretation about these two phenomena is as follows: the filter bubble effect is stronger when users with different preferences have less in common in their respective consumption. This happens when individual mean consumptions are more spread out (higher inter-user diversity), or when individuals consume

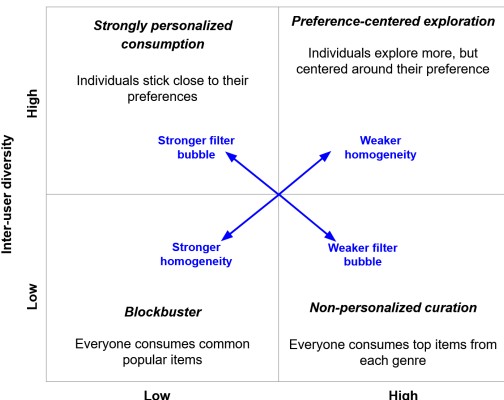

Figure 1: Identifying the strength of homogeneity and filter bubble effect in each of the four possible cases for different levels of inter-user and intra-user diversity. The dynamics seen here motivates our novel definitions of homogeneity and the filter bubble effect.

from a narrower genre range (lower intra-user diversity). On the other hand, homogenization is stronger when users with different preferences are consuming more similar items. This happens when individual mean consumptions are very close (lower inter-user diversity), and individuals consume from a narrower genre range (lower intra-user diversity. In particular, the four scenarios arising from low or high inter-user and intra-user diversity, as well how homogenization and the filter bubble effect change, are presented concisely in figure 1.

Motivated by these observations, we propose the following novel definitions of the filter bubble effect and homogeneization:

**Definition.** *The **filter bubble effect** is given by the ratio between inter-user and intra-user diversity, i.e.,*

$$\textbf{Filter bubble effect} = \frac{\textbf{Inter-user diversity}}{\textbf{Intra-user diversity}} = \frac{\mathrm{Var}_{j \in \mathcal{U}}\left[\mu_j\right]}{\mathbb{E}_{j \sim \mathcal{U}}\left[\sigma_j^2\right]}$$

**Definition.** *Homogeneity, or the level of homogenization of content consumption, is given by the inverse of the standard deviation of the collective genre consumption by all users, i.e.,*

$$\textbf{Homogeneity} = \frac{1}{\sqrt{\mathrm{Var}\left[\bigcup_{j \in \mathcal{U}}\{g_i | i \in C_j\}\right]}}$$

Note that this definition of homogeneity does not explicitly contain inter and intra-user diversity. In order to demonstrate and explain our primary contribution, and motivated by the dynamics presented in figure 1, we adopt the following operationalization of homogeneity:

**Definition.**

$$\textbf{Homogeneity} = \frac{1}{\sqrt{\begin{array}{c}\textbf{Inter-user diversity}^2 \\ +\textbf{Intra-user diversity}^2\end{array}}} = \frac{1}{\sqrt{\begin{array}{c}\mathrm{Var}_{j \in \mathcal{U}}\left[\mu_j\right]^2 \\ +\mathbb{E}_{j \sim \mathcal{U}}\left[\sigma_j^2\right]^2\end{array}}}$$

Figure 2 shows this alternative operationalization of homogeneity against the primary definition of homogeneity. As demonstrated in this figure, the two quantities are highly correlated, with a Pearson correlation coefficient of 0.92823057. This justifies our use of this operationalization.

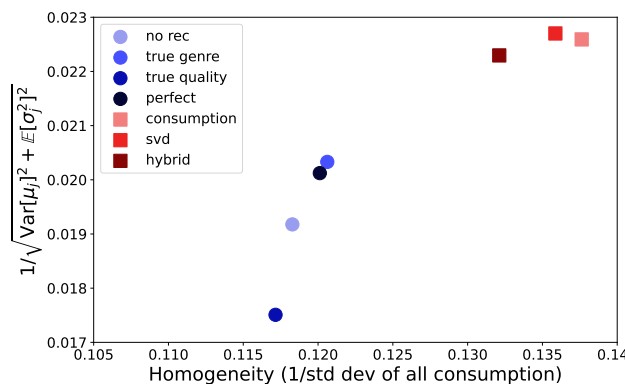

*Figure 2:* $1/\sqrt{inter\text{-}user\ diversity^2 + intra\text{-}user\ diversity^2}$ *against homogeneity (inverse of the standard deviation of all consumption) for each of the seven recommendation algorithms. The Pearson correlation coefficient between the two values is* 0.92823057, *i.e. they are highly correlated.*

## 5 RECOMMENDATION ALGORITHMS

In each round $t$ in a world $\mathcal{W}$ in our simulation framework, the system observes past consumption of users, uses some recommendation algorithm to construct a recommendation $\mathbf{r}_i^j(t)$ from this past data about each existing item $i$ for each user $j$, and sends it to user $j$ in order to guide their consumption choices, e.g., to help user estimation of item utility.

For each item $i$ in world $\mathcal{W}$, we count the number of times it has been consumed at the beginning of round $t$ and denote it via $d_i(t)$. We also define $d_i^{j'}(t) = 1$ if user $j'$ has consumed item $i$ before round $t$ and 0 otherwise.

We test seven recommendation algorithms in our simulation: four act as baselines, while the remaining three are past consumption-based. These algorithms are described below:

**Baseline recommendations:**

(1) **No recommendation:** Our first baseline, where we can observe user consumption patterns without the effects of any recommendation signals. More precisely, $\mathbf{r}_i^j(t) = 0$ for item $i$ and user $j$.
(2) **True genre.** Shows the true genre of an item to users. More precisely, $\mathbf{r}_i^j(t) = (g_i)$ for item $i$ and user $j$.
(3) **True quality.** Shows the true qualities of items to users. More precisely, $\mathbf{r}_i^j(t) = (q_i)$ for item $i$ and user $j$.
(4) **Perfect recommendation:** Shows both the true qualities and the true genres of items to users. More precisely, $\mathbf{r}_i^j(t) = (q_i, g_i)$ for item $i$ and user $j$.

**Past consumption-based recommendations:**

(1) **Consumption:** Shows the number of times an item has been consumed so far to users. More precisely, in this case, $\mathbf{r}_i^j(t) = (d_i(t))$ for item $i$ and user $j$.
(2) **Singular Value Decomposition (SVD):** Shows a weighted version of the consumption number of each item to users. Consumption numbers are weighted by a similarity score between user-user pairs via SVD. More precisely, $\mathbf{r}_i^j(t) =$

$\left(\sum_{j'=1}^m Sim(j, j')d_i^{j'}(t)\right)$ for item $i$ and user $j$. Here, $Sim(j, j')$ is a similarity score between users $j$ and $j'$ computed via SVD.

(3) **Hybrid recommendation:** Shows both the consumption signal and SVD signal of items to users. More precisely, $\mathbf{r}_i^j(t) = \left(d_i(t), \sum_{j'=1}^m Sim(j, j')d_i^{j'}(t)\right)$ for item $i$ and user $j$.

## 6 SIMULATION IMPLEMENTATION

As mentioned in section 3.2.1, we assume that users estimate the utility of each available item from a linear regression model. In order to run our simulation, we need to learn this regression model—we do so by using a simulation process with two phases:

- ***Learning users' utility estimation model.*** In this first phase, we simply learn $w_0$, $\mathbf{w}$ from section 3.2.1. The intuition is that users learn how to interpret and combine different signals about an item and predict its utility from their past consumption experiences. So we simulate user interaction with a similar set of items in order to learn the regression model they come to use to estimate item utility.
- ***Simulating recommendations.*** In this second phase, we simulate the interactions between users and items to generate simulated data about recommendation algorithms and user consumption for our analysis.

Details of how this two-phase simulation process is implemented is provided in appendix C.

## 7 RESULTS

To answer the research question from section 3.1, we examine the simulated data and extract the metrics defined and discussed in section 4. Table 1 describes the specification of the simulation parameters. For each recommendation algorithm discussed here, we run our simulation 15 times with these parameters and report the aggregate results.

Our parameter choices are guided by several assumptions about a realistic user-item interaction. We assume that the number of users is much larger than the number of items (i.e., $m \gg n$). allowing for more information about items and facilitating better learning for algorithms. Given the computational limitations, we choose to use the parameters shown in table 1.

Figure 3 shows inter-user diversity vs. intra-user diversity for various recommendation algorithms. As shown in this figure, past consumption-based recommendations (consumption, SVD and hybrid) induce significantly weaker filter bubble effects compared to no recommendation. In other words, these recommendations alleviate filter bubbles, consistent with previous results. On the other hand, the baseline algorithms that provide accurate genre information (true genre and perfect recommendations) induce stronger filter bubble effects compared to no recommendation.

In particular, this definition of the filter bubble effect is consistent with our general interpretation of the filter bubble effect (i.e. the filter bubble effect is stronger when users with different underlying preferences consume increasingly more different items). We provide empirical evidence for this in appendix A.

| Parameter | Value |
|---|---|
| $\mathcal{Q}$ (Quality) | $\mathcal{N}(\mu_q, \sigma_q^2)$ |
| $\mathcal{G}$ (Item genre) | $\mathcal{N}(0, \sigma_g^2)$ |
| $\mathcal{P}$ (User genre preference) | $\mathcal{N}(0, \sigma_u^2)$ |
| $\mathcal{N}_{qual}$ (Noise in private quality signal) | $\mathcal{N}(0, \sigma_{ps}^2)$ |
| $\mathcal{N}_{genre}$ (Noise in private genre signal) | $\mathcal{N}(0, \sigma_{gs}^2)$ |
| $m$ (Number of users) | 1000 |
| $k_{init}$ (Initial items) | 10 |
| $k_{new}$ (New items per round) | 5 |
| $T$ (Number of rounds) | 100 |
| $k_{train}$ (Training worlds) | 10 |
| $\delta$ (For skewed top pick recommendation) | 1 |
| $k_{top}\%$ (For skewed top pick recommendation) | 25% |

Table 1: Parameter values specifying the simulations reported in this section. Note: While item genres and user preferences are drawn independently from normal distributions for the results reported in this paper, we did replicate these results for bimodal distributions of genres and items. We fixed $\mu_q = 100$, $\sigma_q^2 = \sigma_g^2 = \sigma_u^2 = \sigma_{ps}^2 = \sigma_{qs}^2 = 10$; the results are robust to different values of the mean and standard deviation parameters.

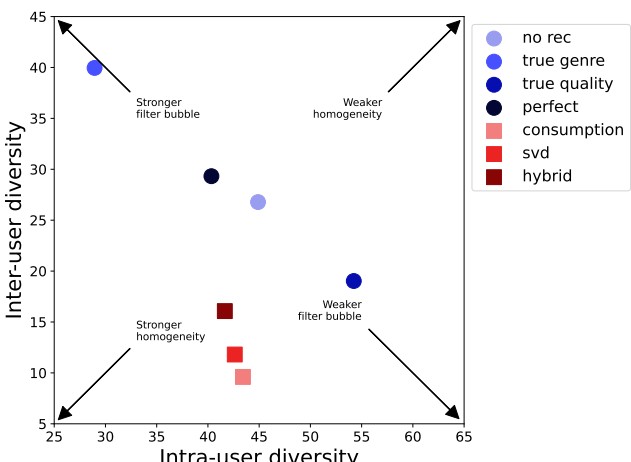

Figure 3: Inter-user diversity vs. intra-user diversity for the recommendation algorithms in section 5. As shown, the baseline algorithms induce a direct trade-off between the two types of diversity. Past consumption-based recommendation algorithms deviate from this trade-off line primarily by reducing inter-user diversity —they do not significantly affect intra-user diversity.

## 7.1 Understanding homogenization-filter bubbles dynamic through effects on diversity

As previously mentioned, our central argument is that to fully understand the role of recommendations in the dynamics between homogenization and the filter bubble effect, we need to examine their impact on both inter-user and intra-user diversity. Therefore, we will now investigate how the algorithms in our simulation affect these two facets of diversity.

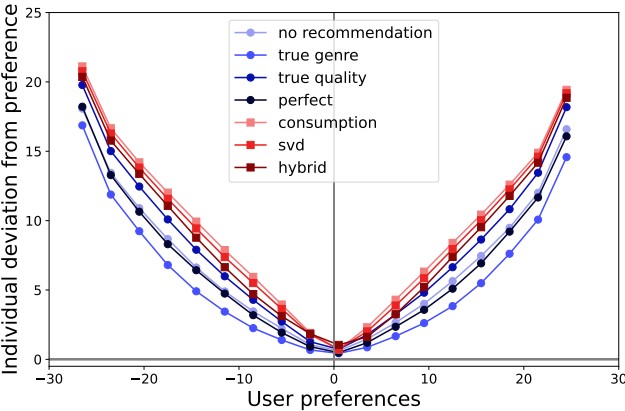

Figure 4: Deviation between mean consumed genre (see section 4) and preference against preferences for the recommendation algorithms in section 5. Compared to no recommendation, past consumption-based recommendations cause large deviations in mean consumed genre towards 0, by pushing all users towards items with near-mode genres.

To do so, we will rely on three key figures. First, figure 3 shows inter-user diversity (Y axis) against intra-user diversity (X axis) for each of the seven algorithms from section 5. Second, figure 4 shows the deviation between preference and mean consumed genre for individual users (Y axis) across varying user preferences (X axis) for all seven recommendation algorithms. Finally, figure 5 shows the consumed genre variance for individual users (Y axis) across varying user preferences (X axis) for all seven recommendation algorithms. For the last two figures, the range of possible user preferences is split into multiple bins, each of size 3. Users from each of the 15 iterations are put into one of these bins. We then report the mean of the relevant statistics for each bin.

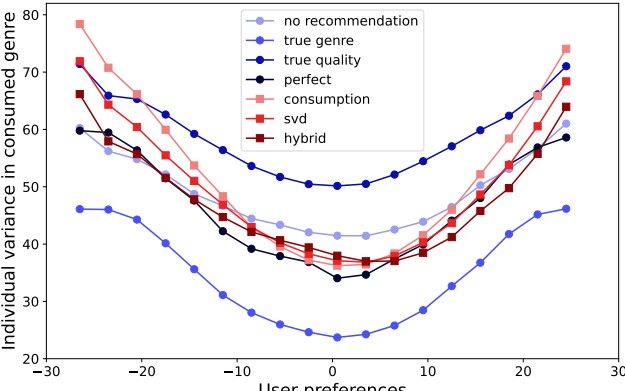

Figure 5: Consumed genre variance (see section 4) against user preferences for the recommendation algorithms in section 5. Compared to no recommendation, past consumption-based recommendations decrease variance for near-mode users and increase variance for niche users by pushing everyone towards blockbuster items.

We use the no recommendation case as our primary baseline. As shown in figure 3, the true genre recommendation achieves higher inter-user diversity and lower intra-user diversity compared to no recommendation, resulting in a stronger filter bubble effect. Without any recommendations, users rely on their personal knowledge

of item qualities and genres. When they have accurate information only about item genres, they prioritize affinity over quality and stick to consuming items closer to their preferences. As a result, they deviate the least from their preferences (figure 4) and consume items from a very narrow genre range (figure 5). We observe similar consumption patterns for perfect recommendation, resulting in a stronger filter bubble effect than no recommendation. When users know both item qualities and genres, they can accurately identify high quality items closer to their preferences and consume those.

On the other hand, true quality recommendation lowers inter-user diversity and increases intra-user diversity compared to no recommendation (figure 3), resulting in a weaker filter bubble effect and weaker homogeneity. When users have accurate knowledge of item qualities but not of item genres, they are more likely to consume high quality items far away from their preferences. As a result, individual mean consumed genre deviates closer to 0 (figure 4), but users consume items from a wider genre range (figure 5).

Meanwhile, past consumption-based recommendations (consumption, SVD, hybrid) rely on past consumption data to learn item quality and are prone to a feedback loop. Since there are more users with near-mode preferences, items with near-mode genres naturally have higher consumption numbers. As a result, these items are favored by past consumption-based recommendation algorithms, which in turn further increases their consumption numbers and continues the loop. These algorithms shift entire user consumption towards the mode of genre distribution rather than widening the range of consumed genres. Therefore, we see large deviations towards 0 in mean consumed genre for niche users (figure 4), while consumed genre variance for individual users do not change much on average compared to no recommendation (figure 5). Consequently, past consumption-based recommendations largely reduce inter-user diversity compared to no recommendation but do not affect intra-user diversity by much (figure 3), resulting in significantly weaker filter bubble effects, and significantly stronger homogeneity.

Combining our observations so far, we can state the following: *past consumption-based recommendations do indeed alleviate filter bubbles, but they do so by greatly reducing inter-user diversity without much effect on intra-user diversity.* Rather than increasing intra-user diversity and exposing users to items from all possible genres, these recommendations primarily shift the consumption of any individual user towards the mode of the genre distribution to increase the similarity in consumption between different users.

## 8 NOVEL RECOMMENDATIONS

Our simulation results demonstrate the importance of considering effects on both inter-user and intra-user diversity when designing recommendation algorithms. Motivated by this insight, next, we propose two novel recommendation ideas aimed at affecting both inter-user and intra-user diversity.

### 8.1 Binned consumption recommendation.

Our first proposed algorithm, binned consumption recommendation, aims to alleviate the filter bubble effect by not only decreasing

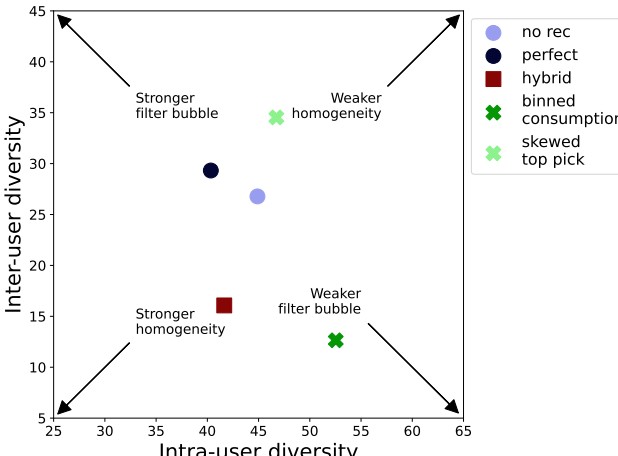

Figure 6: Inter-user diversity vs. intra-user diversity for two novel recommendations, as well as for no recommendation, perfect recommendation and hybrid recommendation. As shown here, binned consumption recommendation greatly increases intra-user diversity and reduces inter-user diversity compared to no recommendation by exposing users towards the popular items from each genre. Skewed top pick recommendation with $\delta = 1$ increases inter-user and intra-user diversity simultaneously compared to no recommendation by exposing near-mode users more towards niche items.

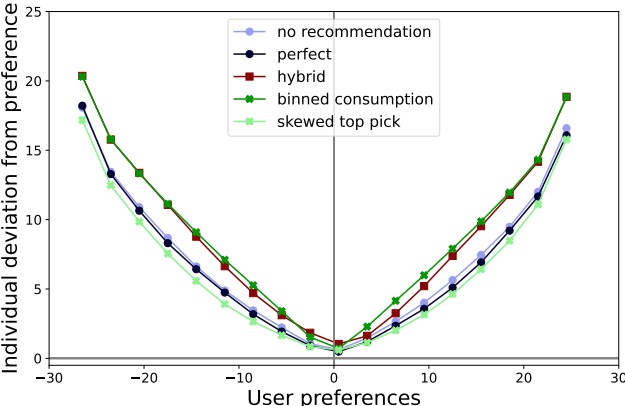

Figure 7: Deviation between mean consumed genre (see section 4) and preference against preferences for two novel recommendations, as well as for no recommendation, perfect recommendation and hybrid recommendation. As shown here, binned consumption recommendation causes deviations similar to past consumption-based recommendation. Since every user consumes the popular items from each genre, everyone's mean consumed genre gets close to 0. Skewed top pick recommendation with $\delta = 1$ causes less deviation compared to no recommendation. It keeps niche users close to their preferences, while exposing near-mode users to niche items from both sides of the mode.

inter-user diversity but also increasing intra-user diversity. It offers non-personalized curation by pushing users towards items with high consumption numbers relative to the rest of their genre. Formally, we define it as:

**Definition** (**Binned consumption recommendation**). The binned consumption recommendation for item $i$ provided to user $j$ is given

by $\mathbf{r}_i^j(t) = \left(\frac{d_i - \mu}{\sigma}\right)$. Here, $\mu$ and $\sigma$ are respectively the mean and the standard deviation of the set $\{d_{i'} | g_{i'} = g_i\}$.

Note that since our model assumes continuous real values for item genre, we discretize the set of possible genre in order to use this recommendation algorithm. The intuition here is to eliminate the implicit bias towards blockbuster items by suppressing the genre information, similar to true quality recommendation.

This algorithm reduces inter-user diversity (figure 6) compared to no recommendation because it nudges all users towards the "popular" items (with high consumption compared to the rest of their genre) and shifts individual mean consumed genre towards 0 (figure 7). However, it increases intra-user diversity compared to no recommendation and past consumption-based recommendations (figure 6) since it helps users consume the popular items and increases the genre range they consume from (figure 8). Finally, with low inter-user diversity and high intra-user diversity, this algorithm significantly weakens the filter bubble effect compared to no recommendation.

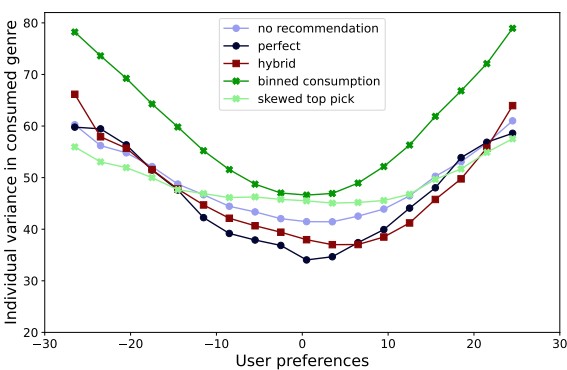

Figure 8: Consumed genre variance (see section 4) against user preferences for two novel recommendations, as well as for no recommendation, perfect recommendation and hybrid recommendation. As shown here, Binned consumption recommendation significantly increases variance for everyone by pushing everyone towards the popular items from each genre. On the other hand, skewed top pick recommendation increases variance for near-mode users by pushing them towards more niche items.

## 8.2 Skewed top pick recommendation.

Our second proposed algorithm, skewed top pick recommendation, focuses on simultaneously increasing inter-user and intra-user diversity rather than alleviating the filter bubble effect. It offers preference-centered exploration to particular groups of users. Formally,

**Definition** (**Skewed top pick recommendation**). First, each item $i$ is ranked according to $\left(q_i^j \cdot |g_i|^\delta\right)$ in descending order. Then, item $i$ is recommended if it is in the top $k_{top}\%$ in this ranking. More precisely, $\mathbf{r}_i^j(t) = 1$ if item $i$ is in the top $k_{top}\%$ of this ranking, and $\mathbf{r}_i^j(t) = 0$ otherwise.

Depending on $\delta$, this recommendation is skewed towards either niche items or items with near-mode genres. With $\delta = 1$, this recommendation algorithm nudges near-mode users more towards niche

items, but does not significantly affect their mean consumed genre since they consume niche items from both sides of the mode. Niche users however stick close to their original preferences. As a result, we see small deviations in mean consumed genre from preferences similar to true genre recommendation (figure 7), and a significant increase in the genre range near-mode users consume from (figure 8). Consequently, this recommendation increases both inter-user and intra-user diversity (figure 6) and causes a stronger filter bubble effect and weaker homogeneity compared to no recommendation.

While not reported here, we did simulate this algorithm for different values of $\delta$. Increasing $\delta$ means that the algorithm will push users more towards niche items. For sufficiently large $\delta$, users begin to ignore the algorithm. As a result, we observe consumption patterns similar to the no recommendation scenario.

## 9 DISCUSSION AND FUTURE DIRECTIONS

We proposed a novel agent-based simulation study to investigate the effects of a select set of recommendation algorithms on user consumption patterns. We developed more refined definitions of the filter bubble and the homogenization effects of recommendations, which decomposed the effect into two components: inter-user diversity and intra-user diversity. Our simulation results show that past consumption-based recommendations only reduce inter-user diversity when alleviating the filter bubble effect —their impact on intra-user diversity is not significant. We then define and examine two novel recommendation algorithms: *binned consumption recommendation*, which provides a non-personalized curated set of content and thus significantly increases intra-user diversity while reducing collective diversity, and *skewed top pick recommendation*, which facilitates preference-centered exploration and thus increases inter-user and intra-user diversity simultaneously.

It should be noted that we do not advocate for either minimizing or maximizing the filter bubble effect; we believe such judgments should be made on a case-by-case basis. Instead of promoting a single ideal approach, our aim is to enable discussions on which strategy is more suitable given a particular context, through our decomposition framework. For instance, a news provider may wish to synchronize user perspectives (reduced filter bubble) while also offering them diverse viewpoints (increased intra-user diversity). As demonstrated by our findings, traditional consumption-based recommendations fall short in this regard.

The scope of our current work leaves ample opportunities for future research. Potential extensions could involve refining our simulation model. Our existing model presupposes that content items are exogenously generated. However, content producers play a crucial role in online ecosystems, influencing the available item pool. Moreover, our model assumes a single, neutral platform, whereas, in practice, multiple platforms, each with distinct objectives, may vie for the attention of users and producers. For example, content creators might migrate to platforms that better serve their genre, leading to genre-specific platforms (e.g., Twitch for live streams, YouTube for long-form videos, TikTok for short clips). Therefore, a logical next step in our research could be to incorporate all three types of agents—users, platforms, and producers—and examine the effects of homogenization and filter bubbles in such a multifaceted ecosystem.

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

## A COMPARISON WITH GENERAL INTERPRETATION OF FILTER BUBBLES

As discussed previously, the general interpretation of the filter bubble effect is that "the filter bubble effect is stronger when people with different underlying preferences consume increasingly more different items". Within our simulation model, we can represent this interpretation by measuring the total pairwise distance in genre between the items consumed respectively by two users on average. The higher this measure is, the stronger the filter bubble effect is for the corresponding recommendation algorithm.

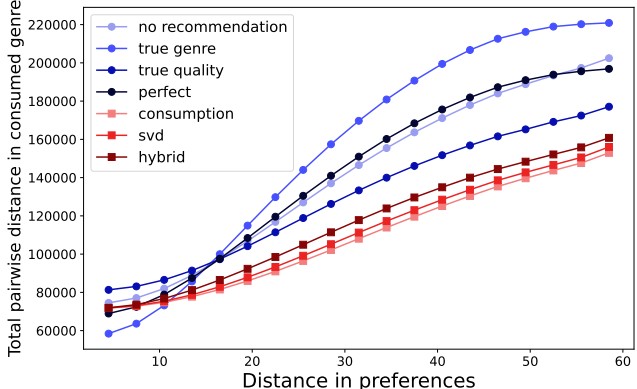

Figure 9: Total pairwise distance in genre between the respective items consumed by two users, vs. the distance between their respective preferences. The higher the position of a curve, the higher the mean total pairwise distance in consumed genre for the corresponding recommendation, and the stronger the filter bubble effect according to the general interpretation.

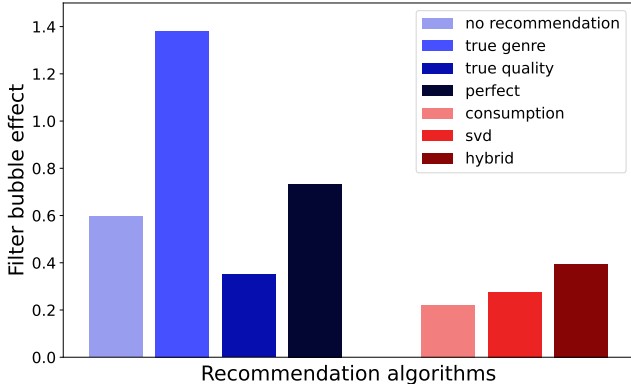

Figure 10: Filter bubble effect (see section 4) for each of the seven recommendation algorithms. As shown here, past consumption-based algorithms significantly decrease the filter bubble effect compared to our baseline algorithms. while true genre recommendation significantly increases it.

Figure 9 shows the total pairwise distance in consumed genre (Y axis) between two users as we vary the distance in preference (X axis) between the users. The range of possible distances in user preferences is split into multiple bins, each of size 3. Every pair of users from each of the 15 iterations are placed into one of these bins based on the distance between them. We then report the mean

total pairwise distance in consumed genre for each bin. Each curve in this figure corresponds to a different recommendation algorithm from section 5, as identified in the legend.

According to the general interpretation, the higher the position of a curve is in figure 9, the stronger the filter bubble effect is for the corresponding recommendation algorithm. This allows us to rank these algorithms in descending order of the strength of the filter bubble effect. We can also rank these algorithms in descending order of our definition of the filter bubble effect (shown in figure 10). We can then verify that the two rankings are exactly the same. This implies that our definition of the filter bubble effect, constructed using inter-user and intra-user diversity, is consistent with the general interpretation of the filter bubble effect.

## B ALTERNATIVE OPERATIONALIZATION OF HOMOGENEITY

In section 4, we provided a definition for homogeneity, and operationalized a measure for homogeneity based on inter and intra-user diversity, namely:

$$\frac{1}{\sqrt{\text{inter-user diversity}^2 + \text{intra-user diversity}^2}}$$

As an alternative, we can also use the following operationalization for this measure:

$$\frac{1}{\text{inter-user diversity} + \text{intra-user diversity}}$$

Figure 11 demonstrates this measure against our definition of homogeneity in section 4. The Pearson correlation coefficient for these two measures is 0.99992089, i.e., they are highly correlated, justifying the use of this measure.

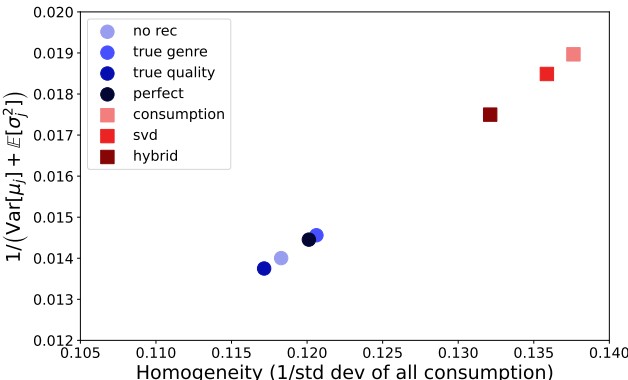

Figure 11: $1/(\text{inter-user diversity} + \text{intra-user diversity})$ against homogeneity (inverse of the standard deviation of all consumption) for each of the seven recommendation algorithms. The Pearson correlation coefficient between the two values is 0.99992089, i.e. they are highly correlated.

## C SIMULATION IMPLEMENTATION DETAILS

In section 6, we briefly described how we leverage a two-phase simulation process to learn the regression model users use to make their decisions. Detailed description of how each of these two phases function are provided below.

## C.1 Learning users' utility estimation model

During the training phase, we initiate $k_{train}$ training worlds $\mathcal{W}^\ell$, $\forall \ell = 1, 2, \ldots, k_{train}$. Each training world simulates the interaction between the users from the true world $\mathcal{W}$ and a set of items similar to those from $\mathcal{W}$. This works as a proxy for the past interactions of users with items, and allows us to learn the regression model used by users in decision making. In particular, we have the following:

- Each training world $\mathcal{W}^\ell$ has the same set of users as $\mathcal{W}$, because we want to learn the regression model used by these particular users.
- Each training world has $n$ items: item qualities and genres in each training world $\mathcal{W}^\ell$ are drawn from the same distributions $Q, \mathcal{G}$ as $\mathcal{W}$. This is because users learn how to interpret various signals about an item (i.e. the regression model) from their past consumption of similar distributions of items.

Therefore, mathematically, each training world can be defined as a collection of $m$ users and necessary distributions:

$$\mathcal{W}^\ell = \left( \{p_1, p_2, \ldots, p_m\}, Q, \mathcal{G}, \mathcal{N}_{qual}, \mathcal{N}_{genre} \right) \quad \forall \ell = 1, 2, \ldots, k_{train}$$

---

**Procedure** Learning phase pseudocode

1  Initialize $\mathcal{W}, \mathcal{W}^\ell \ \forall \ell = 1, 2, \ldots, k_{train}$, each with $k_{init}$ items;

2  Initialize $m$ users, shared across all the simulation worlds;

3  **begin** Training phase

4     **for** *round* $t = 1, 2, \ldots, T$ **do**

5        Construct private signals $q_i^{j\ell}$ and recommendation signals $r_i^{j\ell} \ \forall$ world $\ell$, user $j$, available item $i$ and standardize [2];

6        Construct $X = \left\{ \mathbf{x}_{ji\ell} \right\}_{j,i,\ell}$ where $\mathbf{x}_{ji\ell} = \left[ q_i^{j\ell}, |p_j - g_i^{j\ell}|, r_i^{j\ell} \right]$, $Y = \left\{ U^\ell(j, i) \right\}_{j,i,\ell}$ $\forall$ world $\ell$, user $j$, available item $i$;

7        Learn new $w_0^s, \mathbf{w}^s$ from $(X, Y)$;

8        **for** *training world* $\ell = 1, 2, \ldots, k_{train}$ **do**

9           Add $k_{new}$ items to $\mathcal{W}^\ell$;

10          Construct private signals $q_i^{j\ell}$ and recommendation signals $r_i^{j\ell}$ $\forall$ world $\ell$, user $j$, available item $i$ and standardize;

11          **for** *user* $j = 1, 2, \ldots, m$ **do**

12             User $j$ predicts utility for each available item $i$: $\hat{U}^\ell(j, i) = w_0^s + \mathbf{w}^{s\top} \mathbf{x}_{ji\ell}$ where $\mathbf{x}_{ji\ell} = \left[ q_i^{j\ell}, |p_j - g_i^{j\ell}|, r_i^{j\ell} \right]$;

13             User $j$ chooses unconsumed item $i = \arg\max_i \left\{ \hat{U}^\ell(j, i) \right\}$;

14          Update the consumption numbers of each item $i$;

---

Each training world $\mathcal{W}^\ell$ also has $T$ discrete rounds of progression, similar to our true simulated world $\mathcal{W}$. Each round in a training world progresses similarly to the true simulated world $\mathcal{W}$, except for the following additional mechanism:

At the beginning of each round $s$, consumption data from all $k_{train}$ training worlds is aggregated in order to construct the following training data:

(1) Set $X$ of feature vectors: one feature vector $\mathbf{x}_{ji\ell}$ for each triplet of user $j$, available item $i$ and training world $\mathcal{W}_\ell$. The features in each vector are the signals available to user $j$ about item $i$ in training world $\mathcal{W}_\ell$:

  (a) private quality signal $q_i^{j\ell}$ of user $j$ about the quality of item $i$ in world $\mathcal{W}^\ell$

  (b) perceived distance between the preference of user $j$ and the genre of item $i$ in training world $\mathcal{W}^\ell$, $|p_j - g_i^{j\ell}|$

  (c) recommendation $\mathbf{r}_i^{j\ell}$ about item $i$ for user $j$ in training world $\mathcal{W}^\ell$ provided by the system

Formally, $X = \left\{ \mathbf{x}_{ji\ell} \right\}_{j,i,\ell}$ where $\mathbf{x}_{ji\ell} = \left[ q_i^{j\ell}, |p_j - g_i^{j\ell}|, \mathbf{r}_i^{j\ell} \right]$.

(2) Set $Y$ of target values: the true utility user $j$ would receive from item $i$ in world $\mathcal{W}^\ell$, namely, $U^\ell(j, i) = q_i^\ell - |p_j - g_i^\ell|$.

Formally, $Y = \left\{ U^\ell(j, i) \right\}_{j,i,\ell}$.

From $(X, Y)$, we learn a new regression model (i.e., $w_0^s, \mathbf{w}^s$), which users then use to estimate item utilities and choose their consumption throughout the remainder of round $s$ in each training world. A pseudocode of this phase is provided in Learning phase pseudocode.

## C.2 Simulating recommendations

After the final round of the training phase, we end up with estimates $\hat{w}_0 = w_0^T$, $\hat{\mathbf{w}} = \mathbf{w}^T$ of $w_0, \mathbf{w}$, and are ready to run the simulation in the "true" simulation world $\mathcal{W}$. At each round $s$, each user $j$ estimates the utility of each available item $i$, and chooses to consume the item with the maximum estimated utility. A pseudocode of this phase is provided in Simulation phase pseudocode.

---

**Procedure** Simulation phase pseudocode

1  **begin** Deployment phase

2     **for** *round* $t = 1, 2, \ldots, T$ **do**

3        Add $k_{new}$ items to $\mathcal{W}$;

4        Construct private signals $q_i^j$ $\forall$ user $j$, available item $i$ and standardize;

5        Construct recommendation signals $r_i^j$ $\forall$ user $j$, available item $i$ and standardize;

6        **for** *user* $j = 1, 2, \ldots, m$ **do**

7           User $j$ predicts utility for each available item $i$: $\hat{U}(j, i) = \hat{w}_0 + \hat{\mathbf{w}}^\top \mathbf{x}_{ji}$ where $\mathbf{x}_{ji} = \left[ q_i^j, |p_j - g_i^j|, \mathbf{r}_i^j \right]$;

8           User $j$ chooses unconsumed item $i = \arg\max_i \left\{ \hat{U}(j, i) \right\}$;

9        Update the consumption numbers of each item $i$;

---

