# OpenReview forum: "Disentangling the Long-Term Effects of Recommendations on User Consumption Patterns"
_ACM.org/TheWebConf/2024/Conference — TheWebConf24 Oral_

### Official Review · Reviewer_qAoW · 2023-11-19

**Novelty:** 5
**Technical Quality:** 4

**Review:**

Summary:
The study delves into the homogenization and filter bubble effects within recommendation systems, introducing an agent-based simulation framework to simulate and analyze these phenomena. Two recommendation strategies are proposed based on the observations to enhance diversity awareness and improve the effectiveness of recommendations.

Pros:
1. The research tackles a compelling topic, focusing on the intricate dynamics of homogenization and filter bubble effects in recommender systems.
2. The paper is well-structured, exhibiting clear clarity and easy to follow.

Cons:
1. A notable concern lies in the limited empirical analysis supporting the motivation behind the study and the proposed methodologies. The absence of real-world data to validate the existence of these phenomena and the lack of competing baselines make it challenging to evaluate the effectiveness of the proposed methods. Incorporating empirical evidence and comparative baselines could enhance the robustness of the study.

**Questions:**

See the above drawbacks and please consider including real-world data for model evaluation.

**Reviewer Confidence:**

3: The reviewer is confident but not certain that the evaluation is correct

**Scope:**

4: The work is relevant to the Web and to the track, and is of broad interest to the community

---

### Official Review · Reviewer_mh67 · 2023-11-24

**Novelty:** 5
**Technical Quality:** 6

**Review:**

The paper studies recommendation algorithms and their effects on user choices through a simulation study, using agent based modeling. Specifically, the authors study two important outcomes: homogenization where users consume similar items even though their underlying preferences are different, and filter bubble, where people with different preferences only consume items aligned to their preference. The authors propose studying the above through the following two metrics: inter user diversity and intra user diversity.
Through a simulation study, the authors show that past consumption based recommendations reduce filter bubble effect, and homogenize users towards blockbuster items. Inspired by this, they propose two recommendation strategies. Binned consumption recommendation tries to show a curated set consisting of most consumed items in each genre, to eliminate bias towards blockbuster items. Skewed top pick recommendation on the other hand favors showing more niche items, thereby increasing both inter user and intra user diversity.

The paper is quite well-written and well organized, also the experiments have been performed meticulously. The notation and problems are well-defined.
However, I am not very convinced of the novelty or depth of its contributions and/or conclusions. The experiments though detailed, seems to be purely on simulated or synthetic data, and hence difficult to draw any real world conclusions from them. The theory does not seem very deep. The modeling of users and preferences are all quite standard, and the authors do not apply any novel machine learning (the authors use a linear regression model) and/or algorithmic approaches neither do they study it from a deeper perspective, showing that data drives the conclusions (it is simulations after all). In my opinion, several interesting questions could have been studied with respect to the topics, such as fairness to content, exploration, exploitation, or, if using past consumption information, is changing the projection of users in some latent space in some way, leading to the conclusions.

**Questions:**

1. Could the authors provide some discussion as to how the platform's objective of increasing revenue as well as user engagement etc. interplay with the metrics studied by them?
2. Is there a fairness angle to Skewed top pick recommendation, where the intention is to show more niche items that remain relatively less explored?

**Reviewer Confidence:**

2: The reviewer is willing to defend the evaluation, but it is likely that the reviewer did not understand parts of the paper

**Scope:**

3: The work is somewhat relevant to the Web and to the track, and is of narrow interest to a sub-community

---

### Official Review · Reviewer_qnGm · 2023-11-24

**Novelty:** 4
**Technical Quality:** 3

**Review:**

This paper provides a comprehensive analysis of the impact of recommendation algorithms on user behavior through simulation,  regarding to two critical phenomena: homogenization and filter bubbles.  This paper disentangles the effects of recommendation algorithms on inter-user diversity and intra-user diversity.  The simulations show that traditional recommendation algorithms (based on past behavior) mainly reduce filter bubbles by affecting inter-user diversity without significantly impacting intra-user diversity.  Furthermore, the paper introduces two innovative recommendation algorithms: binned consumption recommendation and skewed top pick recommendation.


My main concern is that the paper lacks an elaboration on the validity of the adopted user preference model and recommendation model, which significantly affects the drawn conclusions.

Additionally, the conclusion drawn in the paper,
>   past consumption-based recommendations do indeed alleviate filter bubbles, but they do so by greatly reducing inter-user
diversity without much effect on intra-user diversity.

appears to contradict previous observations in real-world recommendation systems [1,2]. These observations suggest that users tend to consume increasingly similar items, resulting in low intra-user diversity.  Therefore, it is crucial to engage in further discussions and provide additional analysis regarding this contradiction.

[1] Tien T. Nguyen, Pik-Mai Hui, F. Maxwell Harper, Loren Terveen, and Joseph A. Konstan. 2014. Exploring the Filter Bubble: The Effect of Using Recommender Systems on Content Diversity. In Proceedings of the 23rd International Conference on World Wide Web (Seoul, Korea) (WWW ’14). Association for Computing Machinery, New York, NY, USA, 677–686. https://doi.org/10.1145/2566486.2568012
[2]Zheng, Yu, et al. "DGCN: Diversified recommendation with graph convolutional networks." Proceedings of the Web Conference 2021. 2021.

**Questions:**

Please refer to above section.

**Reviewer Confidence:**

3: The reviewer is confident but not certain that the evaluation is correct

**Scope:**

4: The work is relevant to the Web and to the track, and is of broad interest to the community

---

### Official Review · Reviewer_19hJ · 2023-11-29

**Novelty:** 5
**Technical Quality:** 6

**Review:**

This paper presents a novel approach to analyzing the impact of recommendation algorithms on user consumption patterns, focusing on two key outcomes: homogenization and the filter bubble effect. The study's contribution lies in its redefinition of these concepts, breaking them down into inter-user and intra-user diversity metrics. By employing an agent-based simulation framework, the paper explores how traditional recommendation algorithms based on past behavior predominantly impact inter-user diversity, leading to the development of two new recommendation algorithms aimed at addressing both diversity types.

Strengths:
* The redefinition of homogenization and filter bubble effects and the focus on both inter-user and intra-user diversity is innovative and offers a fresh perspective in the field
* The use of an agent-based simulation to test various recommendation algorithms provides a thorough understanding of the dynamics between homogenization and filter bubble effects

Weaknesses:
* As with these studies, the simulation-based approach, while insightful, might limit the generalizability of the findings to real-world scenarios, as simulations often involve simplifications and assumptions that may not hold in practical environments.

Comments:
* Including a section that applies the findings to real-world data could significantly strengthen the paper’s impact and validate the simulation results though I understand that this is out of scope for this project, it might be something that the authors seriously consider. We have a lot of agent based models which may never be implemented but it might be beneficial for the authors to try to do this.
* Expanding the discussion on the implications of the findings for recommendation system design and user experience in practical settings would be beneficial.

**Questions:**

I do not have any clarifications.

**Reviewer Confidence:**

3: The reviewer is confident but not certain that the evaluation is correct

**Scope:**

4: The work is relevant to the Web and to the track, and is of broad interest to the community

---

### Decision · Program_Chairs · 2024-01-22

**Decision:**

Accept (Oral)

**Comment:**

Summary: The paper analyses impact of recommendation algorithms on user consumption patterns, particularly homogenization and the filter bubble effect.

 Strengths:
 + novel redefinitions of homogenization and filter bubble effects
 + good use of agent-based simulation to test various recommendation algorithms
 + comprehensive analysis via simulation

 Weaknesses:
 - heavy reliance on simulations
 - lack of discussion on real-world scenarios

 Overall, an innovative paper that could use more discussions on practical applications